



# Rapid Landslide Risk Zoning toward Multi-Slope Units of the Neikuihui Tribe for Preliminary Disaster Management

Chih-Chung Chung[1], Zih-Yi Li[2]

[1]Associate Professor, Dept. of Civil Engineering/Research Center for Hazard Mitigation and Prevention, National Central University, 300 Zhongda, Rd., Zhongli Dist., Taoyuan, 320, Taiwan.
[2]Research Assistant, Dept. of Civil Engineering, National Central University, 300 Zhongda, Rd., Zhongli Dist., Taoyuan, 320, Taiwan.

*Correspondence to*: Chih-Chung Chung (ccchung@ncu.edu.tw)

**Abstract.** Taiwan features steep terrain and a fragile geology environment accompanied by frequent earthquakes and typhoons annually. Meanwhile, with the booming economy and rapid population growth, activities are shifting from metropolises to the suburban and mountain areas in Taiwan. However, for example, the Neikuihui tribe in northern Taiwan evolves landslide disasters during extreme rainfall events. To rapidly examine landslide risk in the tribe area for preliminary disaster management, the well-known principle of Risk, which comprises Hazard, Exposure, and Vulnerability, was carefully adapted to examine 14 slope units around the Neikuihui tribe region. The framework of risk zoning is improved based on the previous quantified findings regarding the inventory of the deep-seated landslides in southern Taiwan. Moreover, the proposed procedures comprehensively involve the susceptibility, activity, exposure, and vulnerability of each slope unit. The analyses of the rapid risk zoning toward multi-slope units reveal that No.11 of slope units around the Neikuihui tribe has a relatively higher landslide risk level, and the No.11 slope unit indeed suffered landslide disaster during the typhoon event in 2016. This study preliminarily proves that the proposed framework and details of rapid risk zoning can help identify a relatively high-risk slope unit around a tribal region and address pre-countermeasures for disaster management.

## 1 Introduction

Disasters due to regional landslides, soil yield, and sediment production have received more attention in recent decades (Keefer and Larsen, 2007), and one of the highly correlated causes can be global warming that drastically affects the climate phenomenon, as pointed by Intergovernmental Panel on Climate Change (IPCC) (2014). For instance, rainfall concentrates in fewer and violent rain events in Taiwan, where about 75% of the lands are steep mountains, accompanied by steep terrains, fragile geology conditions, and seismic activities. Therefore, Taiwan suffers several geological disasters due to rainfall events annually. Nevertheless, increasing landslides expand the area of bare land in Taiwan (Chen and Huang, 2010) because of rapid economic development and population growth that push land use to mountainous areas. Taking Typhoon Ophelia in 1990 and Typhoon Herb in 1996, rainfall events caused landslides in Tomong Village (eastern Taiwan) and Nantou (central Taiwan), respectively, and led to costly restoration as well. During Typhoon Morakot in 2009, the mountainous area in southern Taiwan



formed another hot zone for landslides (Lin et al., 2008; 2011; Chen, 2016). Moreover, Nantou's mountainous regions became a spotlight after the Chichi earthquake in 1999 and Typhoon Toraji in 2001 (Lin et al., 2008).

Due to the increase of natural disasters, it is necessary to execute risk management to reduce losses (Chen et al., 2010) and propose an efficient risk assessment to determine where priority must initiate for governance in the case of limited time and
resources (Zheng, 2018). Varnes et al. (1984) revealed a risk assessment principle as Risk = Hazard × Exposure × Vulnerability based on the requests above. Dai et al. (2002) proposed a framework for deep-seated landslide risk assessment, in which Triggering, Preparatory, and Landslide are the primary tasks. Fell et al. (2008) provided guidelines for landslide Susceptibility, Hazard, and Risk zoning for land use planning. Besides, Corominas and Mavrouli (2011) stated that a completed deep-seated landslide risk assessment must include Susceptibility, Hazard, Vulnerability, and Risk. Cantarino et al. (2021) applied the risk
evaluation with Hazard, Exposure, and Vulnerability on expansive residential areas in La Marina, Spain.

In detail, Parise and Wasowski (1999) proposed the areal frequency method to quantify the activity area of a landslide with specified surface features such as cliffs, tension cracks, and slip marks. The equal-area can be identified and drawn from the aerial photos before and after the disaster. Then the active area ratio is defined as the percentage of the active area divided by the total area and used to express the activity of a landslide. Florina (2002) indicated that rock strength, topography (formation
process, slope, and distribution of watersheds), soil, and vegetation are essential factors for distinguishing whether slopes are dangerous or not. Guzzetti (2005) sorted out landslide susceptibility evaluations, in which five methods involve Geomorphological Mapping, Analysis of Landslide Inventories, Heuristic Zoning, Statistical Method, and Deterministic Models. Among these, Statistical Method usually collects numerous landslides to analyze the relationship between slope failure and its factors, such as topographical and geological conditions. Then these factors are weighted and ranked to provide
interpretations for landslide Activity objectively.

Remondo et al. (2005) developed the method to quantitatively assess landslide hazards and risks based on the 140 km$^2$ study area in the lower part of the Deva River Valley, Givascua, Spain. The method incorporated the past landslide inventory frequency and intensity to convert landslide sensitivity into a quantitative hazards model. Remondo et al. (2005) further obtained Vulnerability by quantitatively appraising the damage of each exposed infrastructure, buildings, and land resource.
Comprehensive analysis of landslide Hazard and Vulnerability models can support a quantitative risk model with monetary significance. Di et al. (2008) reported a risk assessment of debris flows in Sichuan Province, China, based on the on-site interpretation from aerial photographs and satellite images. They determined locations of the debris flows and applied GIS to build a database including Hydrology, Topography, Geology, Social and Economic. Regression analysis revealed the relationship between the 24-hour rainfall records and the abovementioned geological and topographic factors. Finally, social
and economic information was jointed to establish a debris flow Vulnerability model, and it was further employed to integrate with debris flow Hazard and Exposure to form a four-stage Risk map.

Fauziah Ahmad et al. (2012) also showed a quantitative risk evaluation method which contains nine environmental risk factors, including Casualties of people, Soil Properties, Earth Coverage, Soil Grading Characteristics, Land Use Suitability, Factor of Safety, Blasting area, Distance between Proposed Structure to Landslide. Then they implemented the method to





examine Penang Island, Malaysia's development area, and results were divided into five levels of Risk: Extremely Low, Low, Medium, High, and Extremely high. After this comprehensive analysis, relevant personnel can use the environmental risk map to measure development feasibility.

To evaluate the susceptibility areas for deep-seated landslides in southern Taiwan, Forestry Bureau initiated a project from 2012 to 2013 at Gaoping River and Zengwen River Basins (He and Lin, 2017). The high-precision digital terrain model (DTM)

surveyed by the LiDAR completed interpretation of deep-seated landslide susceptibility area, and a total of 2,523 places were identified accordingly. In this project, a criterion that a landslide area greater than 10 hectares was defined for a susceptible deep-seated landslide. Then each slope was systematically examined using aerial photographs, hill-shade maps via DTM, and interpretations from various geological and topographic factors. Afterward, susceptibility positions of various deep-seated landslides in the project area were carefully located, and further in-situ inspections were suggested by confirming sliding depth,

local geological survey, and unfavorable hydrological factors. The related products verified the activities of slopes and scale of the landslide dam due to a deep-seated landslide. It is called the evaluation of the occurrence of deep-seated landslide susceptibility. Pan et al. (2019) established a risk assessment framework applied to a deep-seated landslide in Taiwan, including Landslide Susceptibility, Hazard, Vulnerability of protected objects, and Risk Level. They further considered the landslide Activity to reveal its susceptibility to deep-seated landslides. Besides, the Vulnerability of local households, residents, and

infrastructure due to landslide run-out and deposition was also advised. Thus, a deep-seated landslide Risk assessment guide was formed based on the above project in southern Taiwan.

Although the above relevant documents have provided the basic framework required for risk assessment of deep-seated landslides in southern Taiwan, it seems like a pitfall of applying analysis for different landslide types and failure mechanisms (van Westen et al., 2008). Thus, this study refers to the previous framework and aims to provide a rapid landslide risk zoning

based on the improvements, significantly contributing to a smaller scale and multiple slope units around the tribe region. After the comprehensive interpretations of the risk zoning of Neikuihui, the historical disaster event further verifies the feasibility of the proposed method for disaster management.

## 2 Regional details of study area

The administrative area of the Neikuihui tribe belongs to Taoyuan City, as shown in Fig. 1 The total area is around 21 hectares,

and most of the residents live in the northwest of the tribe. It is located on the southern slope of the Yanshan ridge of the Kuihui Mountain, looking to the Ronghua Valley, and two kilometers northwest of Kayilan. There is only a road network to the Neikuihui tribe. Most residents have moved north to the Kuihui Village, and about 15 families live in a small tribe named Neikuihui.

To better visualize the terrain features, the 1 m x 1 m Digital Elevation Model (DEM) is employed as a basement in Fig.2

The overall slope aspect of the tribe is mainly northwest (Fig. 2a), with an average slope degree of 43.9° (Fig. 2b). Fig. 2c and Fig.2d show the CS map and Relief map based on the 1 m x 1 m DEM, respectively. The CSMapMaker plugin on QGIS





produced by Asahi (2014) was applied to generate the CS map, where a topographic map made of altitude, curvature, and slope. (Fig. 2c) is provided with color attributes that Light Blue indicates valleys and Light Red indicates the ridge. Meanwhile, the Relief Visualization Toolbox application (Zakšek et al., 2011; Kokalj et al., 2019) is applied for generating the Relief map

(Fig. 2d) with attributes that the darker the color, the closer the river valley, and the lighter color means the closer to the ridge. These maps can help characterize the slope features quickly.

Figure 2e shows a 1:25,000 geological map from Central Geological Survey (2020) that the strata are the Aoti bottom layer and the Tatongshan layer of interbedded sand and shale. These two layers are apparently conterminous at the Neikuihui tribe. Besides, there is a Ronghua Stream flowing through the Neikuihui tribe region.  Figure 2f also reveals the locations of dip

slopes from the Central Geological Survey (2020), and these dip slopes are geologically sensitive to the safety of the local settlements. This study further details terrain features for the risk zoning based on the previous 1 m x 1 m DEM map, CS map, and Relief map, resulting in eight cliffs and four erosion grooves, as shown in Fig. 2f.

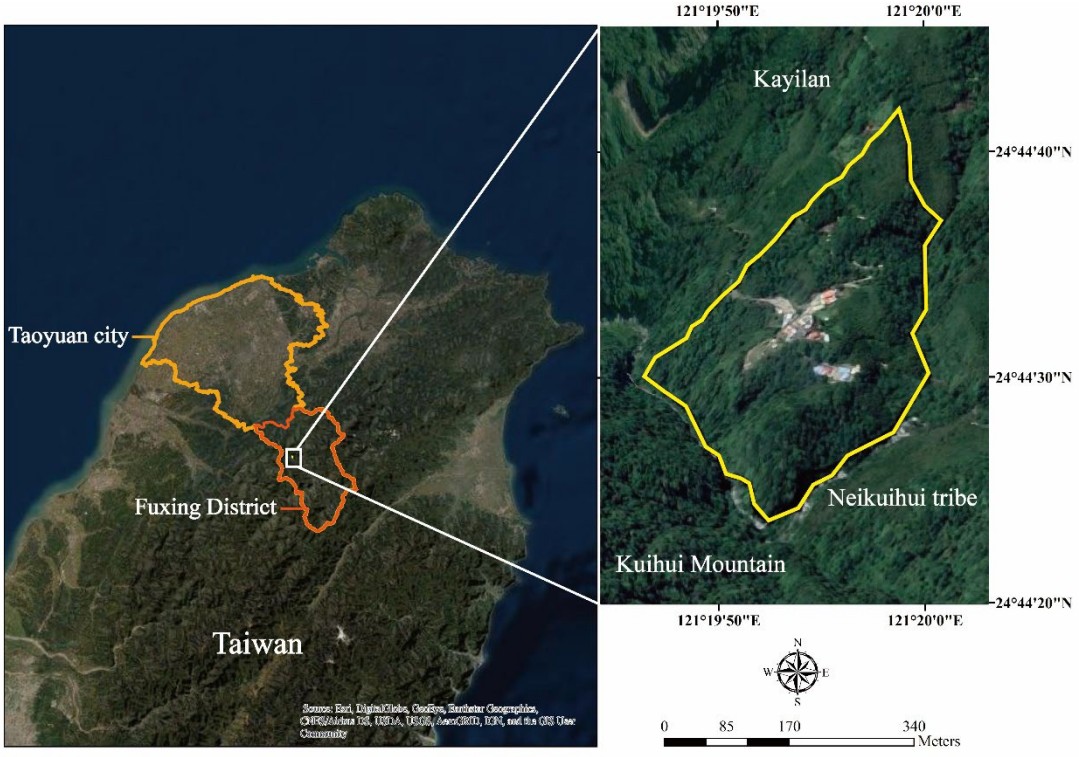

**Figure 1** Location of the Neikuihui tribe (image source: Esri, DigitalGlobe, GeoEye, Earthstar Geographics, CNES/Airbus DS, USDA, USGS, AeroGRID,IGN, and the GIS User Community); Aerial: ESRI ArcGIS 10.4).







**Figure 2** Basic geology and environment of the Neikuihui tribe (**a**) slope aspect (modified after Department of Lands, Ministry of the Interior 2020); (**b**) slope map (modified after Department of Lands, Ministry of the Interior 2020); (**c**) CS map (modified after Department of Lands, Ministry of the Interior 2020); (**d**) Relief map (modified after Department of Lands, Ministry of the Interior 2020); (**e**) 1:250,000 geological map (modified after Central Geological Survey 2020); (**f**) Geologically sensitive area distribution map (modified after Department of Lands, Ministry of the Interior 2020) ; Aerial: ESRI ArcGIS 10.4.



## 3 Improved method for rapid landslide risk zoning

### 3.1 Delimited slope units

Among the processes of delimited slope units, grid-cells units and slope units are commonly adapted (Reichenbach et al., 2018). Usually, grid-cells are directly derived through a DTM or DEM, and the resolution of the predictor variables is assumed as corresponding to that of the DEM pixels, as presented in Fig. 2. Thus, the grid cell division is considered fast and straightforward for better modeling (Van Den Eeckhaut et al., 2009; Rotigliano et al., 2011; Lombardo et al., 2015; Cama et al., 2017). However, the grid unit may not fully express unstable conditions due to its limited range, especially when it is

necessary to predict a slope sliding on a full scale. Nevertheless, pixel-based maps frequently are hard to read and not friendly for land use. To solve the limitation, Martinello et al. (2020) took the Imera Settentrionale watershed in northern Sicily, Italy, as the research scope and found the best way to present the landslide susceptibility map utilizing slope units.

Since a slope unit has more geomorphological and geological significance than a grid unit, a modified method is introduced to delimit slope units and depict slope profiles based on high-resolution DEM (1 m x 1 m) via GIS in this study. The slope-

unit delimiting method is supported with a GIS-based hydrological analysis and modeling tool, Arc Hydro, which originally incorporates DEM and reversed DEM approaches (Maidment, 2002; Xie et al., 2003; Wang et al., 2016). Based on Xie et al. (2004) classification, GIS-based hydrological analysis and modeling tools are implemented to divide the watershed into slope units afterward. In order to make the division accuracy of risk assessment results consistent, Xiong et al. (2019) used the slope units to assess the impact of landslide risk on the operation of oil and gas long-distance pipelines. Based on the previous

experience, this study completes the processing chart of delimited slope units, as illustrated in Fig. 3.

The 14 delimited slope units around the Neikuihui tribe, as shown in Fig. 4, are determined with the steepest slope profiles based on 1 m × 1 m DEM via Hydro-tool in ArcGIS (Xie et al., 2004). Subsequently, related environmental factors were analyzed, including contour, slope aspect, slope degree, valley, and ridge. Then 14 slope units accompanied with cliffs and eroded gullies are manually drawn based on Fig. 2f. According to the improved framework of risk zoning as proposed in Fig.

5, the 14 slope units were graded regarding landslide Hazard, Exposure, and Vulnerability factors. The corresponding Risk scores of each slope unit were consequently obtained. These scores support revealing the Risk level and a disaster reduction strategy for the slope unit to reduce the impact of the disaster.





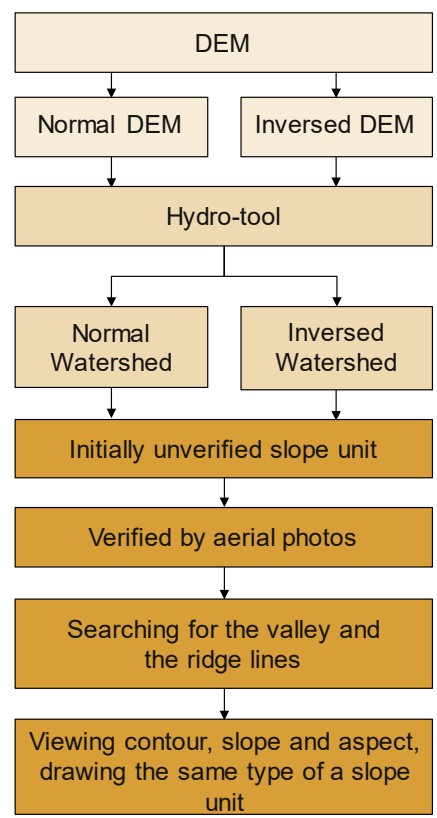

**Figure 3** Process of delimited slope units (modified after Wang et al. 2016).





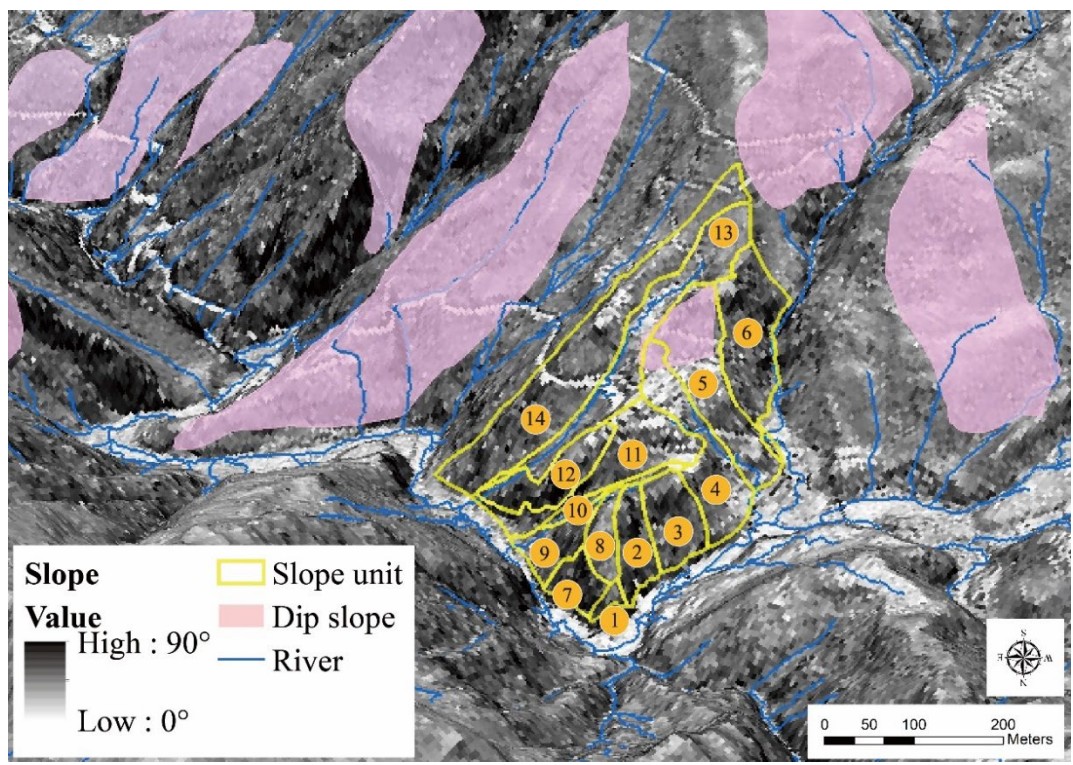

**Figure 4** Distributions of 14 slope units around the Neikuihui tribe (Background: modified after Department of Lands, Ministry of the Interior 2020; Aerial: ESRI ArcGIS 10.4).

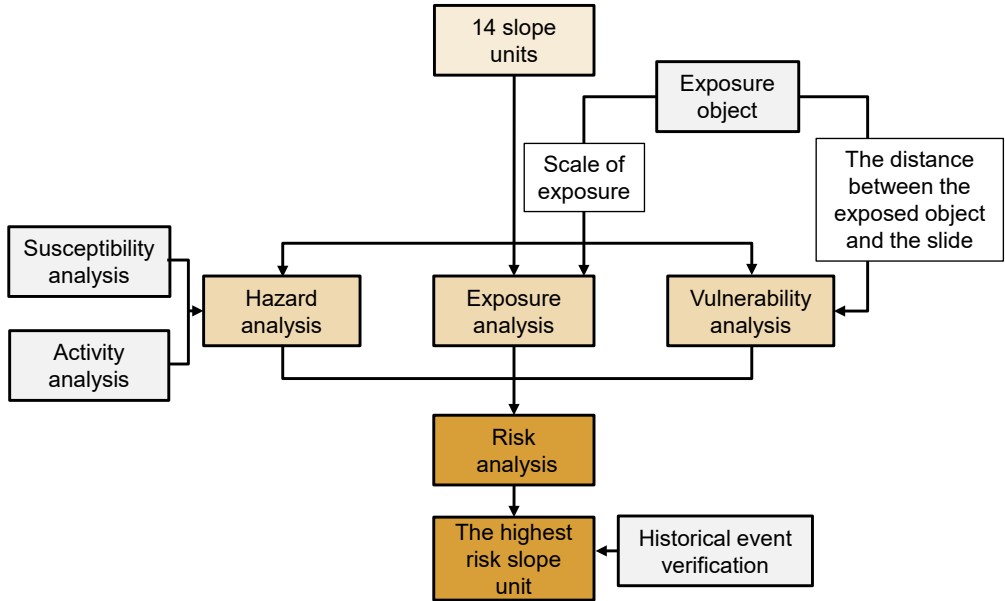

**Figure 5** Improved framework of risk zoning for multi-slope units and the corresponding verification.





## 3.2 Hazard analysis

This study refers to the Hazard of a landslide by considering the indexes of Susceptibility and Activity to identify a landslide in spatial distribution. Although more indexes can be involved for Hazard analysis, the proposed simplified analysis only required the necessary information for rapid risk zoning as described following.

### 3.2.1 Susceptibility analysis

Guzzetti (2005) sorted out the evaluation of the Susceptibility of a landslide, which can be classified into five methods: Geomorphological Mapping, Analysis of Inventories, Heuristic zoning, Statistical Methods, and Deterministic Models. The statistical one uses numerous landslides to analyze the corresponding unstable slopes. Factors such as topographical and geological conditions were marked with weights and rankings statistically, leading to objective results in practice. Forest Bureau (2013) in Taiwan further reported the related results of the river basins in southern Taiwan by including 2523 landslide areas. Then Susceptibility of the landslide was evaluated by weighting factors of the slope degree and distance of river channel, lithology, and dip slope. All factors were proceeded in ArcGIS using the 5-meter grid-size DEM and the improved 1/5000 Gaopingxi watershed geological maps. Afterward, regional statistical results were applied to calculate the occurrence index that was normalized from 1 to 2. Zheng (2018) also proposed the Susceptibility assessment qualitatively and quantitatively by considering the types of past regional landslide events, geological conditions, slope, and aspect.

This study adopts the previous suggestions to classify the Susceptibility of landslides with factors and the corresponding grades. Noticeably, this study designs higher grades of slope degree larger than 45º and river cross or adjacent the slope unit based on the previous experience. Consequently, the grades are further accumulated to evaluate the Susceptibility level of the landslide, as listed in Table 1.

### 3.2.2 Activity analysis

The Activity of a slope can be obtained with long-term monitoring by examining evolutions of slopes through aerial photos of different periods. Parise and Wasowski (1999) proposed Activity Area Ratio (AAR) to quantify the Activity of a slope. AAR was defined as the percentage of the active area to the total area, and the active area often contains specified features such as cliffs, tension cracks, and sliding traces. These features can usually be drawn from the aerial photos before and after the slide events. Base on the aforementioned findings, this study applied the 1.5 m × 1.5 m orthophoto map and Google Earth aerial photos to identify the features mentioned above in each small slope unit around the Neikuihui tribe.

Furthermore, by referring to dip sliding and colluvium indexes proposed by Forest Bureau (2017), the dip sliding indexes include cliff activity, slope toe activity, and the relationship between rock layer orientation and slope aspect/degree. Meanwhile, the colluvium indexes were proposed, including cliff activity, eroded gully activity, and surface features. The criteria of both dip slope (Activity 1) and colluvium (Activity 2) are qualitatively modified in Table 2, respectively. Then, the appraisal of





activity level by considering Activity 1 and Activity 2 is listed in Table 2. Finally, a comprehensive activity score by integrating

levels of Activity 1 and Activity 2 is proposed in Table 3.

**Table 1** Susceptibility grades of environmental factors and Susceptibility level of landslide (modified after Forest Bureau 2013).

| Factor | Classification | Occurrence index | Grades |
|---|---|---|---|
| Slope | >45° | 2 | 4 |
| | 30~45° | 1.5 | 1 |
| | <30° | 1 | 0 |
| River | Cross or adjacent | 2 | 3 |
| | No cross or adjacent | 1 | 1 |
| Lithology | Slate | 2 | 3 |
| | Sandstone, metamorphic sandstone, schist | 1.5 | 2 |
| | Shale | 1 | 1 |
| Dip slope | Yes | 2 | 2 |
| | No | 1 | 1 |
| Fault | Cross or adjacent | 2 | 2 |
| | No cross or adjacent | 1 | 1 |
| Summation of grades | 10~14 | 7~9 | 4~6 |
| Susceptibility level | High | Medium | Low |


### 3.2.3 Hazard level analysis

By considering landslide Susceptibility and Activity level, the combined evaluation for Hazard level is subsequently listed in the upper part of Table 4. The Hazard levels are divided into five classes, then the corresponding Hazard score are listed in Table 4.

**3.3 Exposure analysis**

Among the elements of Risk zoning, an exposed object is significant. For example, a slope is in the mountains with no roads and no households, indicating no damage even if a landslide occurs (Zheng, 2018). Thus, the degree of exposure can refer to the items suffering the slide slope. Finding out and classifying exposed items within the different slopes around the tribe region is crucial in this study. In other words, the degree of damage caused by the impact of a landslide should be critically quantified.

This study referred to the report by Forest Bureau (2017) and redefined the Exposure degree that can be graded according to the exposed objects' importance. The exposed items include affected households in different quantities, the main roads and bridges cross the affected joint, critical public facilities, and reservoir areas. To effectively identify the Exposure level of the protected objects in the tribe region, the number of households of exposed objects is re-adjusted and enhanced, and squared values of the raw grades are presented in Table 5. It also contains the corresponding Exposure score.






**Table 2** Activity grades including dip slope features (referred as Activity 1) and colluvium features (referred as Activity 2) and Activity level (modified after Forest Bureau 2017)

| Activity 1 | | |
|---|---|---|
| Factor | Classification | Grades |
| Cliff activity | The cliff was significantly expanded, tension cracks appeared in the crown, and the back of the deep-seated landslide cliff was eroded. | 3 |
| | The cliff is slightly expanded. | 2 |
| | No significant changes. | 1 |
| Slope toe mobility | The river channel is significantly undercut, causing continuous erosion of the slope toe. | 3 |
| | The river course may erode the slope toe, but the slope toe does not change much. | 2 |
| | No significant changes. | 1 |
| Relationship between rock layer orientation, and slope aspect & degree | Aerial photo interpretation shows that the rock layer is exposed and has the potential of sliding forward. | 3 |
| | Aerial photo interpretation shows that rock layers may be exposed and have the potential to slide forward. | 2 |
| | No significant changes. | 1 |
| Activity 2 | | |
| Factor | Classification | Grades |
| Cliff activity | The cliffs retreat obviously, the scope expands, and the number increases. | 3 |
| | The cliff tends to retreat or expand. | 2 |
| | No significant changes. | 1 |
| Gully activity | Erosion grooves are severely cut down or up, and the number of erosion grooves increases. | 3 |
| | Aerial photos suggest that the erosion ditch may be eroded. | 2 |
| | No significant changes. | 1 |
| Surface features | No vegetation on the surface and exposed rock plates. | 3 |
| | Inclined trees and scattered vegetation. | 2 |
| | Forest is complete and dense. | 1 |
| Summation of grades | 8~9 | 5~7 | 3~4 |
| Activity level | High | Medium | Low |


**Table 3** Comprehensive Activity level of landslide (Forest Bureau 2017)

| Activity 2 \ Activity 1 | High | Medium | Low |
|---|---|---|---|
| High | High | High | Medium |
| Medium | High | Medium | Low |
| Low | Medium | Low | Low |



**Table 4** Hazard level combined by evaluation of landslide Susceptibility and Activity levels

| Activity \ Susceptibility | High | Medium | Low | | |
|---|---|---|---|---|---|
| High | Extremely high | High | Medium | | |
| Medium | High | Medium | Low | | |
| Low | Medium | Low | Extremely low | | |
| Hazard level | Extremely high | High | Medium | Low | Extremely low |
| Hazard score | 5 | 4 | 3 | 2 | 1 |


**Table 5** Grades of exposed objects, and Exposure level and score (modified after Forest Bureau 2013、2017)

| Category | Item | Raw grades | Adjusted grades |
|---|---|---|---|
| Household | More than 5 households | 6 | 36 |
| | Households 3 to 4 | 5 | 25 |
| | Households 1 to 2 | 4 | 16 |
| | Less than 1 household | 3 | 9 |
| Traffic | Main access roads or bridges | 2 | 4 |
| | Ordinary road | 1 | 1 |
| Public Utilities | Public facility, high-voltage towers, and river barriers related to disaster prevention | 4 | 16 |
| Reservoir area | | 4 | 16 |
| Summation of grades | 36~72 | 12~35 | 1~11 |
| Exposure level | High | Medium | Low |
| Exposure score | 3 | 2 | 1 |

### 3.4 Vulnerability analysis

Vulnerability analysis in this study initially represents the degree of damage of the exposed object by considering the relative
position from the landslide, runout, and deposition area. The closer the distance, the greater the damage and the higher
vulnerability. Moreover, the weighting sometimes is considered to be added according to the attributes of the exposed items.
Thus, Vulnerability index proposed by Papathoma-Köhle et al. (2019) was adopted in this study to evaluate the Vulnerability
of the Neikuihui tribe, and details of the Vulnerability index are defined as followings:

Total Vulnerability Score of Household $VS_R = \sum_{i=1}^{NR}(VL_i \times WR)$ , (1)

Total Vulnerability Score of Public $VS_f = \sum_{i=1}^{NF}(VF_i \times WF)$, (2)

where VL is the distance between the household and a susceptibility landslide, divided into three levels (low, medium, high)
ranging from 1 to 3. The closer the distance, the higher VL. Similarly, VF is the distance between the public facilities and a





susceptibility landslide, divided into three levels (low, medium, high) ranging from 1 to 3. WR means the impact of the potential collapse area on residents, and WF means the impact of the potential collapse area on public facilities. Here, the two

weights are set to be one as a fixed value for the preliminary evaluation. NR is the number of households, and NF is the number of public facilities. Then the Vulnerability Score (VS) of a slope unit is the combination of $VS_R + VS_f$, and total weight $W_{total}$ $= NR \times WR + NF \times WF$, then Vulnerability Index (VI) can be written as:

$$VI = VS/W_{total}, \tag{3}$$

Consequently, the Vulnerability level and score based on Vulnerability Index (VI) are revealed in Table 6.

**3.5 Risk analysis**

As revealed in the Introduction section, Varnes et al. (1984) have defined Risk = Hazard × Exposure × Vulnerability, where Hazard, Exposure and Vulnerability are all described in a qualitative way as mentioned above, and the overall structure of risk assessment can achieve the purpose of mutual comparison (Zheng, 2018). The Risk index (RI) of each slope unit around the tribe refers to the ratio of the Risk score (RS) to the total marks of the score ($RS_{max}$) as formulated following (Pan et al. 2019):

$$RI = \frac{Scores\ of\ Hazard \times Exposure \times Vulnerability}{RS_{max}} = \frac{RS}{RS_{max}}, \tag{4}$$

where the $RS_{max}$ is 75 as the summation of the maximums of previous scores. Then the Risk level of a slope can be obtained from Table 6.

**Table 6** Vulnerability & Risk index, level, and score

| Vulnerability index (VI) | 3≧VI>2.5 | 2.5≧VI>2.0 | 2.0≧VI>1.5 | 1.5≧VI≧1.0 | 1.0≧VI≧0 |
|---|---|---|---|---|---|
| Vulnerability level | Extremely high | High | Medium | Low | Extremely low |
| Vulnerability score | 5 | 4 | 3 | 2 | 1 |
| Risk index (RI) | 1.0≧RI>0.8 | 0.8≧RI>0.6 | 0.6≧RI>0.4 | 0.4≧RI≧0.2 | RI<0.2 |
| Risk level | Extremely high | High | Medium | Low | Extremely low |


**4 Improved method for rapid landslide risk zoning**

**4.1 Hazard analysis results**

**4.1.1 Susceptibility and Activity analysis results**

Based on the classification of the slope, river distance, lithology, dip slope, and distance away from the main geological

structure in Table 1 and basic information as shown in Fig. 2, landslide susceptibility levels of 14 slope units of the Neikuihui





tribe are provided and depicted in Fig. 6, and the corresponding grades are listed in Table 7. Among 14 slope units, No. 1, 2, 3, 6, 7, and 9 have high landslide susceptibility levels where the slopes degrees are above 45° and are adjacent to or intersecting with the river.

Besides, based on the classification of cliffs, slope toes, rock formations, erosion gullies, and surface features as mentioned in Tables 2, this study gives levels of Activity 1 and Activity 2 of 14 slope units around the Neikuihui tribe as detailed in Table 8 and Table 9, respectively. According to the comprehensive activity level as defined in Table 3, because the cliff is slightly expanded and the river channel is significantly undercut and leads to continuous erosion of the slope toe, aerial photo interpretation shows that rock layers would be exposed and have the susceptibility to slide forward. Such that No. 6, 11, 13, and 14 slope units have high Activity levels, as shown in Fig. 7.


**Figure 6** Susceptibility level mapping of landslide referred to slope units around the Neikuihui tribe (Background: modified after Department of Lands, Ministry of the Interior 2020; Aerial: ESRI ArcGIS 10.4).



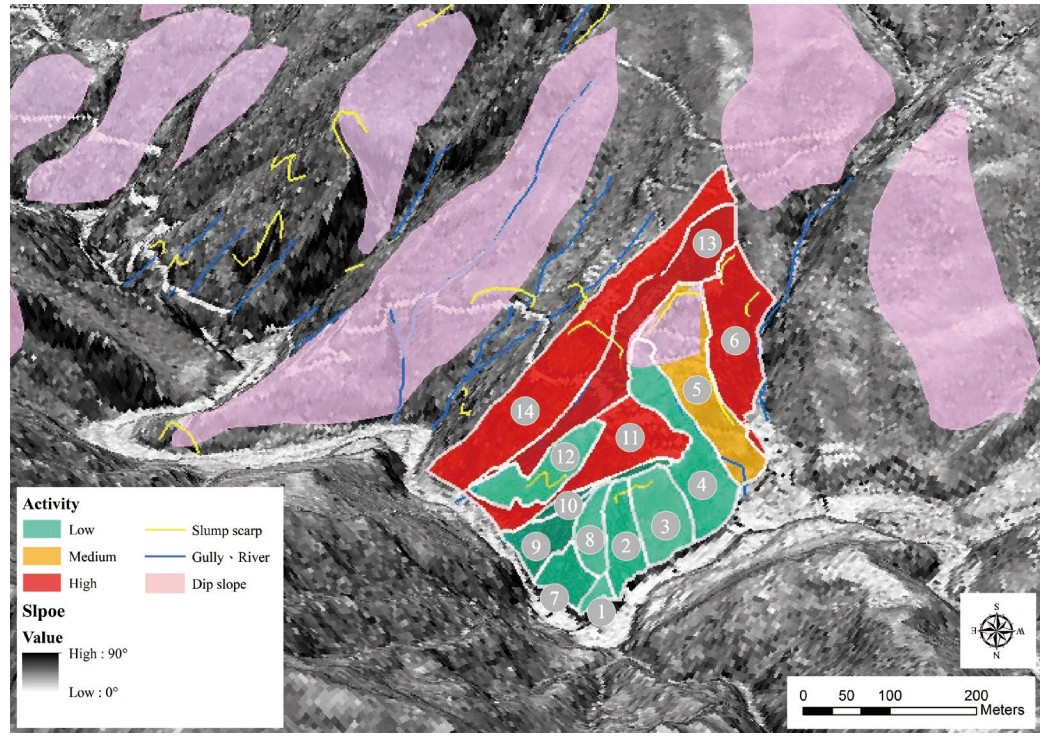

**Figure 7** Activity level mapping of landslide referred to slope units around The Neikuihui tribe (Background: modified after Department of Lands, Ministry of the Interior 2020; Aerial: ESRI ArcGIS 10.4).





**Table 7** Grades and levels of landslide Susceptibility of slope units around the Neikuihui tribe

| Slope unit | Slope Degree | Slope Grade | River cross or adjacent Attributes | River cross or adjacent Grade | Lithology Attributes | Lithology Grade | Dip slope Attributes | Dip slope Grade | Fault cross or adjacent Attributes | Fault cross or adjacent Grade | Total grades | Susceptibility level |
|---|---|---|---|---|---|---|---|---|---|---|---|---|
| No.1 | 63.2° | 4 | Yes | 3 | | 2 | No | 1 | | 1 | 11 | High |
| No.2 | 54.0° | 4 | Yes | 3 | | 2 | No | 1 | | 1 | 11 | High |
| No.3 | 47.1° | 4 | Yes | 3 | | 2 | No | 1 | | 1 | 11 | High |
| No.4 | 32.7° | 1 | Yes | 3 | | 2 | Yes | 2 | | 1 | 9 | Medium |
| No.5 | 31.8° | 1 | Yes | 3 | | 2 | Yes | 2 | | 1 | 9 | Medium |
| No.6 | 46.5° | 4 | Yes | 3 | Sandstone, | 2 | No | 1 | | 1 | 11 | High |
| No.7 | 61.2° | 4 | Yes | 3 | metamorphic | 2 | No | 1 | | 1 | 11 | High |
| No.8 | 37.4° | 1 | No | 1 | sandstone, | 2 | No | 1 | No | 1 | 6 | Low |
| No.9 | 47.6° | 4 | Yes | 3 | schist | 2 | No | 1 | | 1 | 11 | High |
| No.10 | 39.0° | 1 | Yes | 3 | | 2 | No | 1 | | 1 | 8 | Medium |
| No.11 | 31.1° | 1 | Yes | 3 | | 2 | No | 1 | | 1 | 8 | Medium |
| No.12 | 55.4° | 4 | No | 1 | | 2 | No | 1 | | 1 | 9 | Medium |
| No.13 | 30.3° | 1 | Yes | 3 | | 2 | No | 1 | | 1 | 8 | Medium |
| No.14 | 37.6° | 1 | Yes | 3 | | 2 | No | 1 | | 1 | 8 | Medium |



265                **Table 8** Grades and levels of Activity 1 of slope unit around the Neikuihui tribe.

| Slope unit | Cliff activity1 | | Slope toe mobility | | Relationship between rock layer orientation, and slope | | Total grades | Activity1 level |
|---|---|---|---|---|---|---|---|---|
| | Attributes | Grade | Attributes | Grade | Attributes | Grade | | |
| No.1 | No significant changes | 1 | The river course may erode the slope toe | 2 | No significant changes | 1 | 4 | Low |
| No.2 | | 1 | | 2 | | 1 | 4 | |
| No.3 | | 1 | | 2 | | 1 | 4 | |
| No.4 | | 1 | | 2 | The rock layer may be exposed and slippery | 2 | 5 | Medium |
| No.5 | Slightly expanded | 2 | | 2 | | 2 | 6 | |
| No.6 | Significantly expanded | 3 | The river channel is significantly undercut | 3 | | 2 | 8 | High |
| No.7 | No significant changes | 1 | No significant changes | 1 | No significant changes | 1 | 3 | Low |
| No.8 | | 1 | | 1 | | 1 | 3 | |
| No.9 | | 1 | | 1 | | 1 | 3 | |
| No.10 | | 1 | The river course may erode the slope toe | 2 | | 1 | 4 | |
| No.11 | Significantly expanded | 3 | The river channel is significantly undercut | 3 | The rock layer may be exposed and slippery | 2 | 8 | High |
| No.12 | No significant changes | 1 | No significant changes | 1 | No significant changes | 1 | 3 | Low |
| No.13 | Significantly expanded | 3 | The river channel is significantly undercut | 3 | The rock layer may be exposed and slippery | 2 | 8 | High |
| No.14 | | 3 | | 3 | | 2 | 8 | |


**Table 9** Corresponding grades and levels of Activity 2 of slope units around the Neikuihui tribe.

| Slope unit | Cliff activity2 Attributes | Grade | Gully activity Attributes | Grade | Surface features Attributes | Grade | Total grades | Activity2 level | Comprehensive Activity level |
|---|---|---|---|---|---|---|---|---|---|
| No.1 | No significant changes | 1 | The erosion ditch may be eroded | 2 | Forest is complete and dense | 1 | 4 | | |
| No.2 | | 1 | | 2 | | 1 | 4 | Low | Low |
| No.3 | | 1 | | 2 | | 1 | 4 | | |
| No.4 | | 1 | | 2 | | 1 | 4 | | |
| No.5 | The cliff tends to retreat or expand | 2 | | 2 | Inclined trees and scattered vegetation | 2 | 6 | Medium | Medium |
| No.6 | | 2 | Erosion grooves are severely cut down or up | 3 | | 2 | 7 | | High |
| No.7 | No significant changes | 1 | No significant changes | 1 | Forest is complete and dense | 1 | 3 | | |
| No.8 | | 1 | | 1 | | 1 | 3 | | |
| No.9 | | 1 | | 1 | | 1 | 3 | Low | Low |
| No.10 | | 1 | The erosion ditch may be eroded | 2 | | 1 | 4 | | |
| No.11 | The cliffs retreat obviously | 3 | Erosion grooves are severely cut down or up | 3 | Inclined trees and scattered vegetation | 2 | 8 | High | High |
| No.12 | No significant changes | 1 | No significant changes | 1 | Forest is complete and dense | 1 | 3 | Low | Low |
| No.13 | The cliffs retreat obviously | 3 | Erosion grooves are severely cut down or up | 3 | Inclined trees and scattered vegetation | 2 | 8 | High | High |
| No.14 | | 3 | | 3 | | 2 | 8 | | |





### 4.1.2 Hazard level and score results

By combining the evaluation results of landslide Susceptibility and Activity levels as referring to Table 4, Hazard scores of each slope unit can be obtained as listed in Table 4, leading to analyzed results of landslide Hazard levels of 14 slope units around the Neikuihui tribe as detailed in Table 10, as well as the corresponding illustration map as depicted in Fig. 8. The No. 6 slope unit has an Extremely High level of landslide hazard, and No.11, 13, and 14 have High levels of landslide hazard. It can be reasoned that No.6 is next to the river channel and significantly undercut, as shown in Fig. 8.

### 4.2 Exposure analysis results

Based on the classification of households, transportation, and essential facilities as listed in Table 5, this study determined Exposure levels of 14 slope units around the Neikuihui tribe as detailed in Table 11, and the mapping as shown in Fig. 9, which is based on the aerial photo to visualize the resident locations better. Results indicate that No. 4, 5, 6, 11, 13, and 14 slope units have higher Exposure levels. Noticeably, the residents of the Nekuihui tribe mainly live in the No. 4 and No. 11 slope units. Besides, landslides may block the major external road leading to the outside, causing evacuation and material transportation difficulties. Consequently, these two units have the highest Exposure levels.

**Table 10** Hazard levels and scores of slope units around the Neikuihui tribe

| Slope unit | Susceptibility Level | Activity level | Hazard Level | Hazard Score |
|---|---|---|---|---|
| No.1 | High | Low | Medium | 3 |
| No.2 | | | Medium | 3 |
| No.3 | | | | 3 |
| No.4 | Medium | Low | Low | 2 |
| No.5 | | Medium | Medium | 3 |
| No.6 | High | High | Extremely high | 5 |
| No.7 | | Low | Medium | 3 |
| No.8 | Low | | Extremely low | 1 |
| No.9 | High | | Medium | 3 |
| No.10 | Medium | | Low | 2 |
| No.11 | | High | High | 4 |
| No.12 | | Low | Low | 2 |
| No.13 | | High | High | 4 |
| No.14 | | | | 4 |





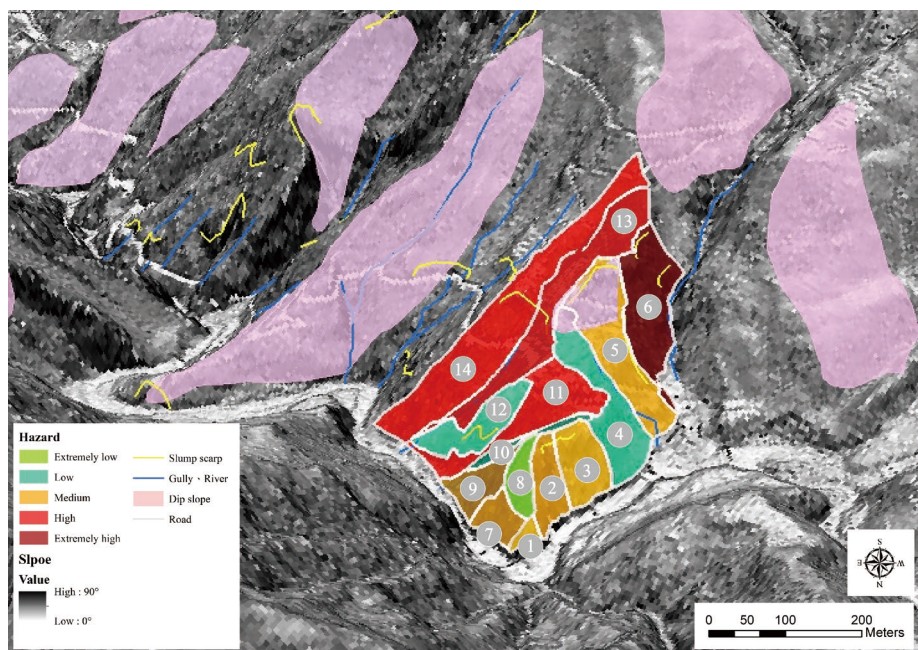

**Figure 8** Hazard level mapping of landslide referred to slope units around the Neikuihui tribe (Background: modified after

Department of Lands, Ministry of the Interior 2020; Aerial: ESRI ArcGIS 10.4).


**Table 11** Exposure levels and scores of slope units around the Neikuihui tribe.

| Slope unit | Household | | Traffic | | Public utilities | | Total grades | Exposure level | Exposure score |
|---|---|---|---|---|---|---|---|---|---|
| | Attributes | Grade | Attributes | Grade | Attributes | Grade | | | |
| No.1 | | 9 | | 0 | | 0 | 9 | | 1 |
| No.2 | < 1 household | 9 | No | 0 | | 0 | 9 | Low | 1 |
| No.3 | | 9 | | 0 | | 0 | 9 | | 1 |
| No.4 | > 5 households | 36 | Main access roads or bridges | 4 | | 0 | 40 | High | 3 |
| No.5 | Households 1 to 2 | 16 | | 4 | | 0 | 20 | Medium | 2 |
| No.6 | | 9 | | 4 | | 0 | 13 | | 2 |
| No.7 | | 9 | | 0 | | 0 | 9 | | 1 |
| No.8 | < 1 household | 9 | No | 0 | No | 0 | 9 | Low | 1 |
| No.9 | | 9 | | 0 | | 0 | 9 | | 1 |
| No.10 | | 9 | | 0 | | 0 | 9 | | 1 |
| No.11 | > 5 households | 36 | Main access roads or bridges | 4 | | 0 | 40 | High | 3 |
| No.12 | | 9 | No | 0 | | 0 | 9 | Low | 1 |
| No.13 | < 1 household | 9 | Main access roads or bridges | 4 | | 0 | 13 | Medium | 2 |
| No.14 | | 9 | Main access roads or bridges | 4 | | 0 | 13 | | 2 |

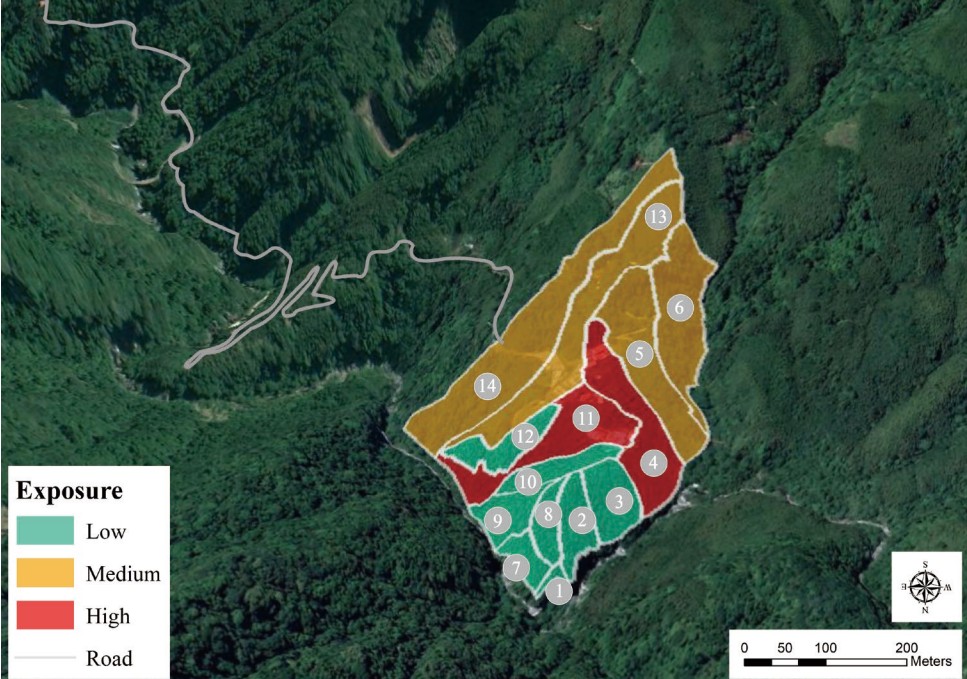

**Figure 9** Exposure level mapping of landslide referred to units around the Neikuihui tribe (Background: modified after © Google Earth 2021; Aerial: ESRI ArcGIS 10.4).


## 4.3 Vulnerability analysis results

With the description of the Vulnerability assessment method in the previous section, this study used 1 m × 1 m DEM to manually interpret the cliffs and erosion ditches around the Neikuihui tribe, as shown in Fig. 2f. Zheng (2018) proposed that the possible impact range of the landslide is interpreted based on the principle of the landslide area and the farthest run-out

distance, and the transport of landslides will not be blocked by the terrain if there is no high or steep terrain. Besides, the landslide transport's distance and direction are kept in the origin state, indicating that the influence range would cover the source area and landslide path. However, the path of the landslide should not exceed the ridgeline of the slope. After entering the flat ground, landslide gradually spreads, and deposits accumulate. According to the above principles, this study judges the possible impact range of the soil and rock if the susceptibility collapse area collapses. The calculation process of the

Vulnerability index proposed by Papathoma-Köhle et al. (2019) is applied to analyze the Vulnerability level of 14 slope units around the Neikuihui tribe. The relevant results are listed in Table 12. For example, $W_{total}$ of the No. 5 slope unit equals 9, and VS = 3 × (3 × 1) + 6 × (1 × 1) = 15. Thus, the Vulnerability index of the No. 5 slope unit is 15 / 9 = 1.67. Accordingly, No. 4, 5, 11, 13, and 14 slope units can be found with apparent Vulnerability levels because these slope units have residents, as shown in Fig. 10. Compared with other slope units, the No.11 slope unit has the highest Vulnerability level because the residents



living on slope unit No. 11 are the closest to the susceptibility collapse and are within the range that the collapsed soil and rock

may cover.

**Table 12** Vulnerability rating of slope units around the Neikuihui tribe.

| Slope unit | Household | VL WR=1 High (3) | Medium (2) | Low (1) | Total Vulnerability grades of household | Public facilities | VL WF=1 High (3) | Medium (2) | Low (1) | Total Vulnerability grades of public utilities | Sum of W | Total Vulnerability score | Vulnerability index | Vulnerability level | Vulnerability score |
|---|---|---|---|---|---|---|---|---|---|---|---|---|---|---|---|
| No.1 | 0 | 0 | 0 | 0 | 0 | No | 0 | 0 | 0 | 0 | 0 | 0 | 0 | Extremely low | 1 |
| No.2 | 0 | 0 | 0 | 0 | 0 | No | 0 | 0 | 0 | 0 | 0 | 0 | 0 | | 1 |
| No.3 | 0 | 0 | 0 | 0 | 0 | No | 0 | 0 | 0 | 0 | 0 | 0 | 0 | | 1 |
| No.4 | 9 | 3 | 0 | 6 | 15 | No | 0 | 0 | 0 | 0 | 9 | 15 | 1.67 | Medium | 3 |
| No.5 | 9 | 3 | 0 | 6 | 15 | No | 0 | 0 | 0 | 0 | 9 | 15 | 1.67 | | 3 |
| No.6 | 0 | 0 | 0 | 0 | 0 | No | 0 | 0 | 0 | 0 | 0 | 0 | 0 | | 1 |
| No.7 | 0 | 0 | 0 | 0 | 0 | No | 0 | 0 | 0 | 0 | 0 | 0 | 0 | Extremely low | 1 |
| No.8 | 0 | 0 | 0 | 0 | 0 | No | 0 | 0 | 0 | 0 | 0 | 0 | 0 | | 1 |
| No.9 | 0 | 0 | 0 | 0 | 0 | No | 0 | 0 | 0 | 0 | 0 | 0 | 0 | | 1 |
| No.10 | 0 | 0 | 0 | 0 | 0 | No | 0 | 0 | 0 | 0 | 0 | 0 | 0 | | 1 |
| No.11 | 5 | 5 | 0 | 0 | 15 | No | 0 | 0 | 0 | 0 | 5 | 15 | 3 | Extremely high | 5 |
| No.12 | 0 | 0 | 0 | 0 | 0 | No | 0 | 0 | 0 | 0 | 0 | 0 | 0 | Extremely low | 1 |
| No.13 | 1 | 0 | 0 | 1 | 1 | No | 0 | 0 | 0 | 0 | 1 | 1 | 1 | Low | 2 |
| No.14 | 1 | 0 | 0 | 1 | 1 | No | 0 | 0 | 0 | 0 | 1 | 1 | 1 | | 2 |






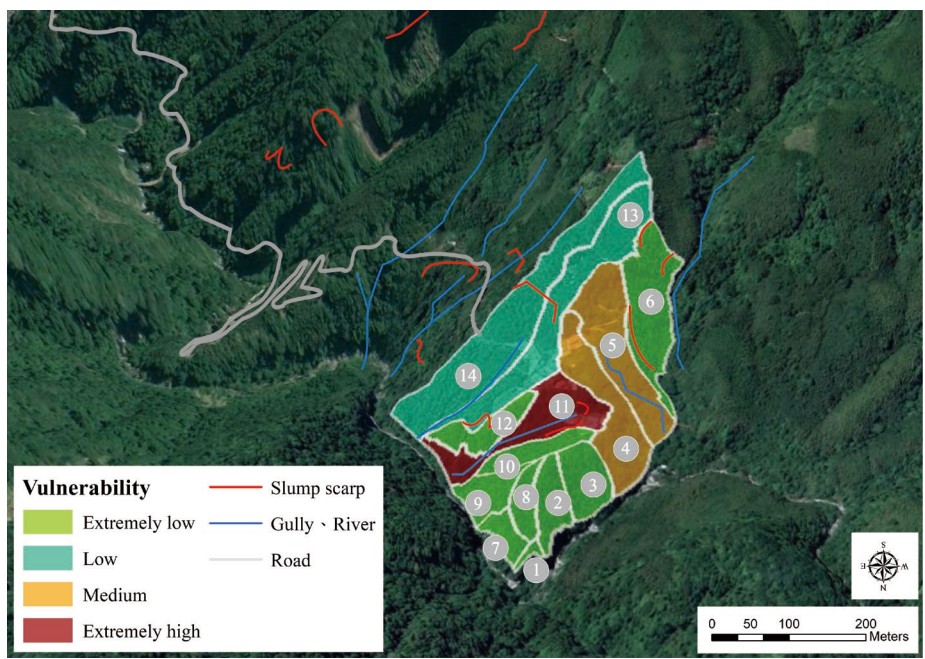

**Figure 10** Vulnerability level mapping of landslide referred to units around the Neikuihui tribe (Background: modified after modified after © Google Earth 2021; Aerial: ESRI ArcGIS 10.4).

**4.4 Risk analysis results**

By combining the scores of Hazard, Exposure, and Vulnerability of each slop unit around the Neikuihui tribe, Risk levels were then obtained by score summation as listed in Table 13. The No.11 slope unit is High of Risk level, and No. 4, 5, 13, and 14 slope units are Low of the Risk level. The corresponding mapping result of the landslide Risk is shown in Fig 11. Since slope unit No. 6 is steep and tangent to the river, it is judged that its hazard level is the highest as processing hazard analysis at the beginning. However, no one living there, and the landslide has few impacts on human lives and public facilities nearby. Thus,

the Risk level is not as high as the slope unit No. 11. Through these three main elements to do Risk zoning toward slope units, we can quickly identify which areas need to be prioritized in disaster prevention.





**Table 13** Risk levels of slope units around the Neikuihui tribe

| Slope unit | Hazard score | Exposure score | Vulnerability score | Risk score | Risk index | Risk level |
|---|---|---|---|---|---|---|
| No.1 | 3 | 1 | 1 | 3 | 0.04 | |
| No.2 | 3 | 1 | 1 | 3 | 0.04 | Extremely low |
| No.3 | 3 | 1 | 1 | 3 | 0.04 | |
| No.4 | 2 | 3 | 3 | 18 | 0.24 | Low |
| No.5 | 3 | 2 | 3 | 18 | 0.24 | |
| No.6 | 5 | 2 | 1 | 10 | 0.13 | |
| No.7 | 3 | 1 | 1 | 3 | 0.04 | |
| No.8 | 1 | 1 | 1 | 1 | 0.01 | Extremely low |
| No.9 | 3 | 1 | 1 | 3 | 0.04 | |
| No.10 | 2 | 1 | 1 | 2 | 0.02 | |
| No.11 | 4 | 3 | 5 | 60 | 0.80 | High |
| No.12 | 2 | 1 | 1 | 2 | 0.02 | Extremely low |
| No.13 | 4 | 2 | 2 | 16 | 0.21 | Low |
| No.14 | 4 | 2 | 2 | 16 | 0.21 | |


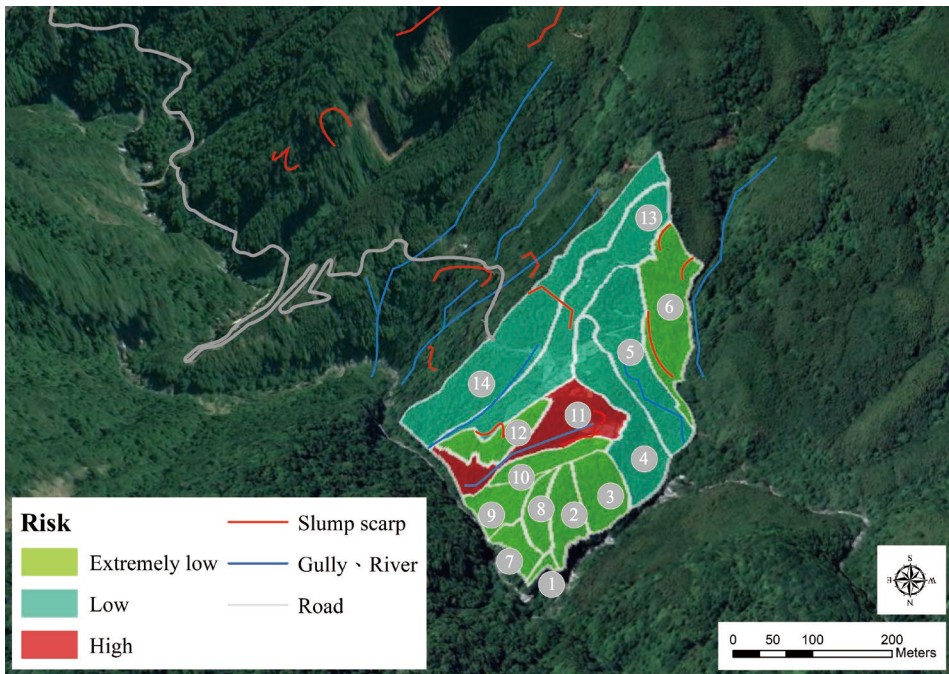

**Figure 11** Risk level mapping of landslide referred to slope units around the Neikuihui tribe (Background: modified

after © Google Earth 2021; Aerial: ESRI ArcGIS 10.4).





## 5 Discussions

Since the Risk level of the No.11 slope unit is High, as illustrated in Fig. 11, this study further compared the results with historical disaster events for validation. According to the 2016 report (Bureau of Soil and Water Conservation of the Agriculture), the No. 11 slope unit area was affected by the torrential rainfall during Typhoon Meiji on September 27, 2016. A landslide disaster occurred at 14:00 on the same day. Referring to the historical data of the Fuxing Rain Gauge Station closest to the disaster location, the hour rainfall record at this time was 45.5 mm, as depicted in Fig. 12. The road's foundation

was scoured by rainwater, resulting in a landslide with a length of 8-10 meters and a depth of 30-60 cm, as shown in Fig. 13a. The soil yields moved down and rushed into the No. 8 residential house (Fig. 13b), and trees and telephone poles were seriously inclined at the No. 11 slope unit (Fig. 13c). Through the evidence of landslide disaster at No. 11 slope unit, the rapid risk zoning method toward multi slope units around the tribal region is proven preliminarily.

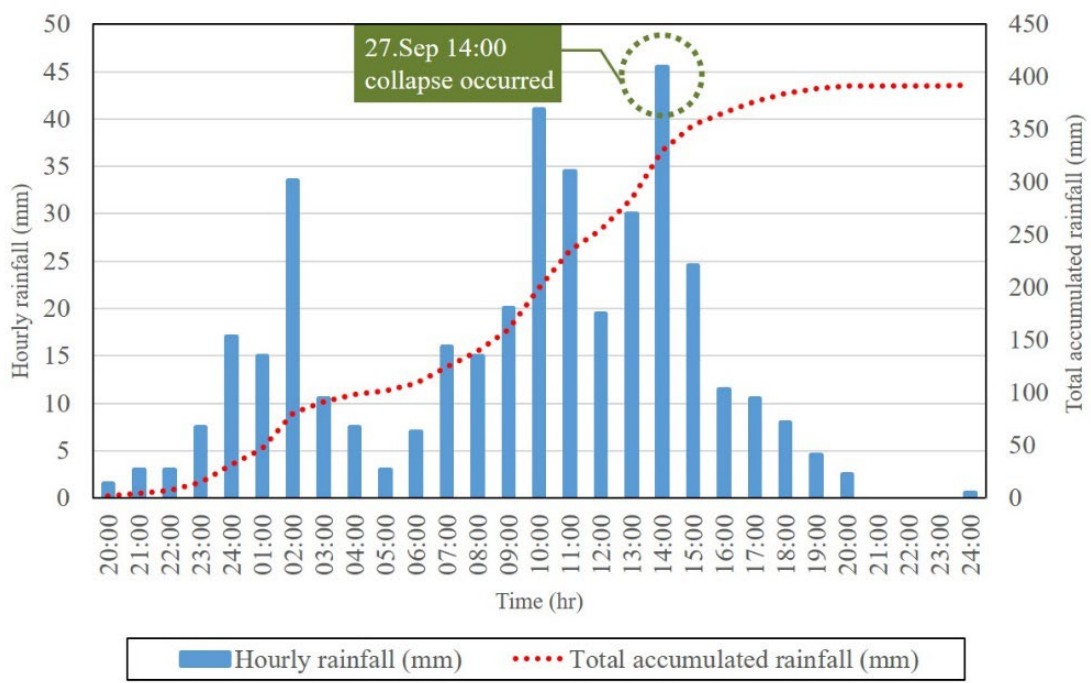


**Figure 12** Rainfall records during Typhoon Meiji (September 26 to September 27, 2016).




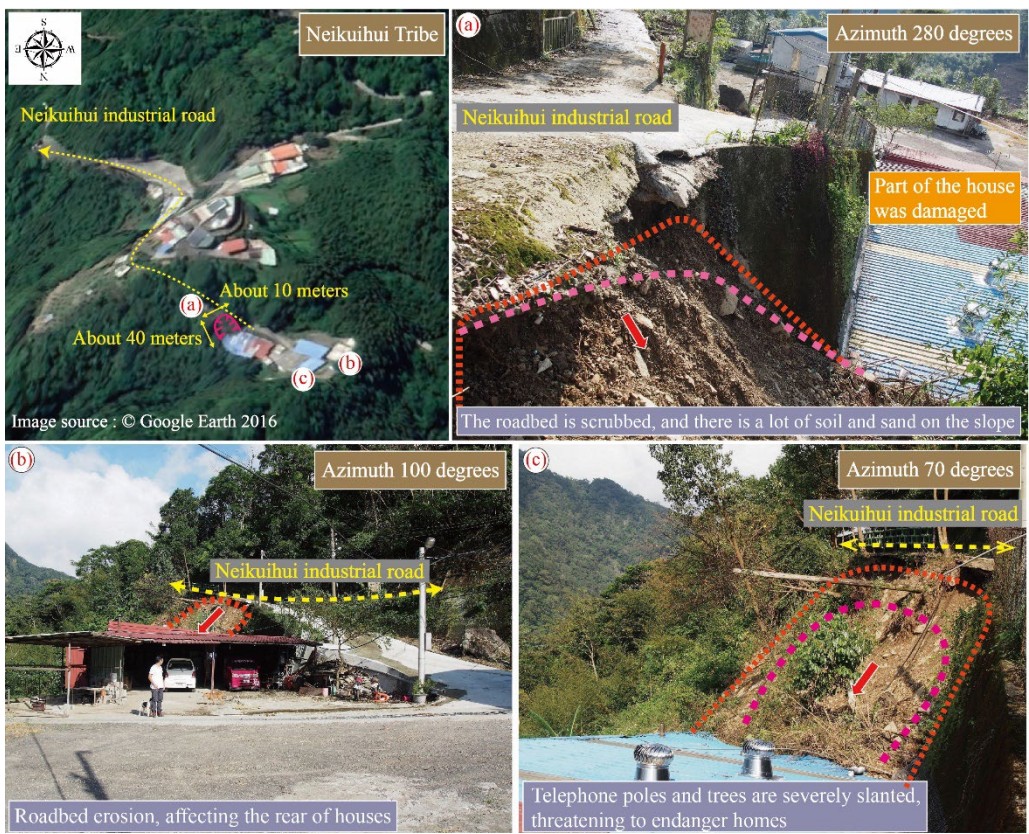

**Figure 13** Historical disaster event at No.11 slope unit during Typhoon Meiji 2016 (Picture source: 2016 Water and Soil Conservation Bureau; Aerial: ESRI ArcGIS 10.4).


The primary purpose of this research is to establish a simple risk assessment framework for quickly interpreting the collapse of multi-slope units in settlements. Through the primary concepts referred to previous relevant findings, the proposed method can quickly assess the risk of slope collapse in various regions. The advantages and limitations summarized by this research are as follows:

1.  The proposed Risk zoning of collapse covers the assessments of Hazard, Exposure, and Vulnerability, in which both the object of preservation and its value are carefully considered. If the Risk zoning is implemented in a large-scale collapse, the above concepts are still necessary accompanied by extended methods. For example, the effect of the more prolonged runout distance of the very large-scale landslide should be re-considered in terms of semi-experienced statistical prediction (Zheng, 2018). This study recommends refining Risk zoning with a variety of data integrity and alternative

360        methods.

2.  Risk zoning can be expressed in qualitative or quantitative methods. The quantitative method is introduced, but it requires very detailed site information and statistics of various parameters, often taking a lot of time and cost. A qualitative





description does not need to quantify each factor, which is described hierarchically. Although the qualitative method is less accurate, it can initially manifest the differences and ranks of the various sites, which is helpful to quickly provide a reference for subsequent risk management. Therefore, the initial risk assessment is more suitable with qualitative descriptions.

3. The Risk zoning framework designed in this study include the Activity assessment when grading the Hazard, which can make up for the lack of time change in the Susceptibility assessment and represent the actual site activities.

4. At present, this research assumes that one-time mass destruction will occur in the susceptibility area. However, some susceptibility areas may be damaged by local erosion and erosion repeatably. If different types of damage (corrosion or falling rocks) can be classified in the future, resulting in a complete risk zoning.

5. After conducting Risk zoning, those slopes with higher Risk levels can be subsequently evaluated by straightforward methods, such as on-site surveys, geological drilling, and numerical simulations, and the results can be practiced as a reference for further governance.

## 6 Conclusions

Due to Taiwan's steep terrain and fragile geology, coupled with the frequent occurrences of typhoons and earthquakes, tribes at mountainous areas accompanied by rapid economic development and activity might evolve into landslide disasters. This study draws up a framework of rapid Risk zoning toward multi-slope units around the tribal region and integrates qualitative/quantitative concepts with landslide Susceptibility, Activity, Exposure, and Vulnerability. Then the Neikuihui tribe in northern Taiwan was taken as an example for validation.

Research results indicated that the No. 11 slope unit has a high landslide Risk level. At the same time, it is verified by a historical disaster event, indicating that the modified Risk zoning is feasible for multi-slope units around the tribal region. The proposed procedure can benefit government agencies for rapidly conducting preliminary analysis of the risk of tribe regions and prioritizing the disaster mitigation countermeasures. For example, by revealing the risk zoning for residents as a reference, the government can guide residents to voluntarily inspect the community environment and plan evacuation during the disaster. More importantly, the government can strengthen disaster prevention and relief awareness, and promotes regular emergency disaster relief drills at the usual time. This modified process of Risk zoning is further suggested to be validated comprehensively with other cases of tribe regions.

## Data availability

The information referring to risk map (in shp files) can be obtained from fromhttps://www.dropbox.com/sh/r5z0ecxqm5tmtsd/AAAM8wnDeVY5ZTigAMgWCh39a?dl=0. Further information can be made available upon request to the corresponding author.



**Author contributions**

All authors contributed to conceptualisation, led by CCC, who also conducted the formal analysis and initial draft. JSPJ had a leading role on risk zoning perspective. ZYL contributed to validation and data visualisation. CCC critically reviewed the paper and contributed to the preparation of the final version.

**Competing interests**

The authors declare that they have no conflict of interest.

**Acknowledgements**

This work was supported by the Ministry of Science and Technology (MOST) of Taiwan under grant number MOST 109-2119-M-008-005.

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
