# Peer review of "Rapid Landslide Risk Zoning toward Multi-Slope Units of the Neikuihui Tribe for Preliminary Disaster Management"

_Natural Hazards and Earth System Sciences, 2021_

## Author Response (AR1)

Paper Title: " Rapid Landslide Risk Zoning toward Multi-Slope Units around the Neikuihui Tribe for Preliminary Disaster Management "

**Reviewer #1:**

| Comment | Author's Response |
|---|---|
| The manuscript describes a rapid landslide risk zoning approach targeted at supporting preliminary disaster management in Taiwan. Overall, this is clearly an application-oriented work, and scientific novelty is somewhat limited. However, since the geological disasters do occur quite frequently in this area, entailing that also case-study-type studies aiming at effective disaster management is of importance, I think that this topic is generally suitable to be published in NHESS. | Authors appreciate the comment. This study aims to provide a framework of Risk zoning which comprehensively applies the Susceptibility, Activity, Exposure, and Vulnerability of each slope unit. The scientific novelty may be limited, but the proposed assessment is practical to quickly identify a relatively high-risk slope unit around a tribal region and address pre-countermeasures for disaster management. |
| 1.  In general, I would term this indicator-based approach as a semi-quantitative method rather than a fully quantitative one throughout the manuscript. Indicators and weights are not set on an fully quantitative basis, but rather by using expert knowledge. Susceptibility models based on statistical learning or vulnerability curves based on exact measurements of damage and process intensity would be fully quantitative. Many concepts used here imply equal weights or nearly linear relationships on ordinal scales (e.g. "low - medium - high"). This is not a criticism of the | Thank you for the suggestion, and authors agree the term of "indicator-based". As mentioned in the manuscript, the evaluation indicators of Susceptibility in Table 1 are based on the previous logistic result of 2523 collapses in the southern Taiwan, leading to the major influence factors, such as lithology, slope degree, elevation, and dip slope. The authors then modified the results by using expert knowledge to establish a susceptibility grades fitted to the rapid landslide risk zoning toward a tribe area. |

| | | |
|---|---|---|
| | method per se, | |
| 2. | Abstract: The information about the "No.11 slope unit" is too specific to be mentioned in the abstract without further context. This particular slope unit will be unknown to a vast majority of readers. | Thank you for suggestion. The Abstract content is modified as: "The rapid risk zoning analysis of multi-slope units delivers a sloping unit with a high level of landslide risk, and this slope unit did suffer from landslide disasters in the 2016 typhoon event." |
| 3. | Figure 2: It would be helpful to include the variable displayed in each facet, e.g. in the legend for quicker understanding. | Thank you for suggestion, legends for c and d in Fig.2 have been added for quick understanding. |
| 4. | l. 120: "Among the processes of delimited slope units, grid-cells units and slope units are commonly adapted (Reichenbach et al., 2018)." Personally, I find the beginning of this section to be a bit sudden. The concept of slope units should be briefly described here, including methods for delineating them. This would aid the reader, especially those who are not familiar with the concept. | Thank you for suggestion. The content is modified with new citations as: Quantitative geomorphological and environmental analysis requires the adoption of well–defined spatial domains as basic mapping units. The spatial domains provide local boundaries to aggregate environmental and morphometric variables for related analyses (Alvioli et al.,2020). Grid cells and slope units are commonly adapted among the spatial domain processes of delimited slope units (Reichenbach et al., 2018). Grid cells, typically aligned with a digital elevation model, are the standard mapping unit preference (Alvioli et al.,2020). Usually, grid cells are directly derived through a DTM or DEM, and the resolution of the predictor variables is assumed as |

corresponding to that of the DEM pixels. Therefore, the grid cell division is considered fast and straightforward for modeling (Van Den Eeckhaut et al., 2009; Rotigliano et al., 2011; Lombardo et al., 2015; Cama et al., 2017). Despite its popularity and operational advantages, grid cells have apparent drawbacks for susceptibility modeling (Guzzetti et al., 1999). First, there is no physical relationship between landslides and a grid cell or a group of grid cells since landslides from slope processes acting at different spatial and temporal scales result in geomorphological forms of very different shapes and sizes (Malamud et al., 2004; Guzzetti et al., 2012). An alternative to grid cells is the method of slope units, which refers to hydrological terrain divisions bounded by drainage and ridges (Carrara, 1983; Carrara et al., 1991, Carrara et al., 1995; Guzzetti et al., 1999). Martinello et al. (2020) brought the Imera Settentrionale watershed in northern Sicily, Italy, as the research scope and found a better way to present the landslide susceptibility map utilizing slope units.

The size of the slope units can be tailored to the type and size of the landslides since a slope unit has more geomorphological and geological significance than a grid unit (Carrara et al., 1991; Alvioli et al., 2016). Accordingly, a modified method is introduced to delimit slope units and depict slope profiles based on high-resolution DEM (1 m x 1 m) via GIS in this study. The slope-unit delimiting method is supported by a GIS-based hydrological analysis and modeling tool,

| | |
|---|---|
| | Arc Hydro, which originally incorporates DEM and reversed DEM approaches (Maidment, 2002; Xie et al., 2003; Wang et al., 2016). Based on Xie et al. (2004) classification, GIS-based hydrological analysis and modeling tools are implemented to divide the watershed into slope units through the proposed processing chart of delimited slope units, as illustrated in Fig. 3. |
| 5. Table 1 mixes two types of information. Suggest to separate the lower part of the table to keep the datasets tidy; otherwise it might be confusing why "Classification" would correspond to 10~14; "Occurrence index" would correspond to 7~9, and "Grades" would correspond to "4~6". This is obviously not meant here. | Thank you for suggestion, the authors have modified all tables as individual ones accordingly. |
| 6. The same applies to Table 2, Table 4 and Table 5. | Thank you for suggestion, the authors have modified all tables accordingly. |
| 7. 3.2.1 Susceptibility analysis: The exact method used is unclear to me. l. 160 describes that "Susceptibility of the landslide was evaluated by weighting factors of the slope degree and distance of river channel, lithology, and dip slope", which might indicate some sort of (logistic?) regression model. However, this assumption somewhat contradicted by Table 1, which looks more | Thank you for suggestion. The related response can be referred to specified Comment 1, and the sentence is modified as "Then Susceptibility of the landslide was evaluated logistically with main factors of the slope degree, lithology, and dip slope, as well as the adjacent conditions to a river and fault." |

| | | |
|---|---|---|
| | like an indicator-based approach (which would be more in line with the rest of the described approach)? Please clarify. | |
| 8. | l. 191: "Among the elements of Risk zoning, an exposed object is significant." Please clarify/rephrase this sentence. | Thank you for suggestion. The sentence is modified as "It is essential to calculate how many households, traffic, and public utilities are exposed to risk zoning". |
| 9. | 3.4 Vulnerability analysis: "Vulnerability analysis in this study initially represents the degree of damage of the exposed object by considering the relative position from the landslide, runout, and deposition area. The closer the distance, the greater the damage and the higher vulnerability." I think that the assumption that vulnerability (denoted as damage in this context) dependes on the distance to the landslide is too simplistic. Many factors do determine physical vulnerability (c.f. https://doi.org/10.1016/j.jhydrol.2022.127501), and even deposition height might not be a fully adequate indicator of physical vulnerability, let alone distance. While I understand why this was done due to practical reasons (lack of better information), I think this should be openly discussed as a limitation. | Thank you for suggestion. Authors agree that vulnerability depending on the distance to the landslide is simplistic. However, the primary purpose of this study is to establish a rapid risk assessment framework for quickly interpreting the landslide of multi-slope units in a tribe area. We have added a description to 3.4 section and also the study limitations to the Discussion as follows: "In order to quickly assess the Vulnerability, it may be simplistic but efficient to judge the vulnerability score by considering the possible impact area of the landslide and the distance from the household / public facilities with the limited geological and geomorphological data. However, there are still households in this area, and the economic conditions are disadvantaged. According to the developed methodology in this study, when the survey resources are limited, the administration can easily and quickly remind people in higher-risk areas to relocate to a safe place." |

| | |
|---|---|
| 10. Figure 12: I suggest to use a "prettier" functional relationship between the two y-axis, e.g. y2 = 10*y1 (i.e. scale for total accumulated rainfall is [0, 500]). Otherwise, the horizontal lines are off, and the labels for "Total accumulated rainfall" are floating around without corresponding horizontal grid line. | The data interval 0-500 mm for total accumulated rainfall in Fig.13 has been corrected as suggested. |
| 11. Discussion: I think that the pros and cons of using an indicator-based approach like this one could be discussed in more detail, including the setup of the single elements (tables). | Based on the above suggestions, the authors have supplemented the details in the discussion, and we take No. 11 slope unit as an example to reveal the step of the rapid Risk zoning. |
| 12. Discussion: Validation is performed basically with one event on Slope No. 11 during a Typhoon event. This is ok, but n=1 is more anecdotal evidence rather than a convincing sample size for accurate validation of the approach. Some sort of goodness-of-fit metric would be good. Since this can probably not be achieved within the scope of this study, accuracy could be discussed based on plausibility and local expert knowledge, and validation procedures could be outlined. | Thank you for the suggestion. Since the purpose of this study is to quickly analyze the landslide risk in a region through indicators, it is suggested that after the follow-up assessment of the landslide risk in a certain region, the analysis results should be verified by interviewing residents, experts and scholars for plausibility. |
| 13. Data availability: Please consider depositing the data to a more persistent repository, e.g. the PANGAEA data repository or zenodo. | Thank you for your suggestion, we will look for a database suitable for storage. |

| | |
|---|---|
| 14. Technical corrections: Figure 2 / Figure 8: "Slpoe" (probably incorrect layer name) | Thank you for suggestion, the wrong word has been edited. |

**Reviewer #2:**

| Comment | Author's Response |
|---|---|
| My main concern is the evaluation of the hazard. The hazard is evaluated here by combining a susceptibility and two activity criteria. The susceptibility analysis is based a method developed for deep-seated landslides (unfortunately only published in Chinese) defined as having a surface of at least 10 hectares but is applied here on a region that is around 20 hectares according to the text and about 10 hectares according to my measurements on Figure 1. In addition, the only example of landslide that is given is a landslide with a depth of 30-60 cm, so I doubt that the method developed for deep-seated landslides can be used here. It would be necessary to better describe the type of landslide that is considered to make sure that the method is appropriate. When it comes to the activity criteria, I do not really understand:

    (1). why it is divided in two when there is a lot of redundancy between the two grades.
    (2). why the relation between the bedding and the slope is considered in the activity in addition to dip-slope being a | (1). The authors appreciate the suggestion. Susceptibility is typically applied for the landslide risk assessment of large-scale geological conditions accompanied by common environmental factors, such as slope degree and lithology. To rapidly assess the landslide risk of a tribe region, this study refers to the susceptibility findings of deep-seated landslide inventory and carefully includes Activity analysis, especially for a small-scale slope unit. The revised Activity is based on the Activity Area Ratio (AAR) principle and the categories of dip sliding and colluvium indexes from the previous experience. Hence, the proposed Activity aims to examine the evolutions of slope units through DEM and aerial photos of different periods. Regarding this, Activity 1 (Table 3) is modified to measure the activity level of the dip sliding along a slope unit, while Activity 2 (Table 4) examines the activity level of the colluvium layer on the surface of a slope unit.

(2). The relation between the bedding and the slope considered in the |

| | |
|---|---|
| criterion in the susceptibility analysis.
   (3). what cliff activity means. I assume that it refers to the scarp, but I am a bit confuse here about the landslide type that is considered. I assume those criteria were developed to be applied to evaluate an acceleration of slow-moving landslides.

I would like to see some examples of how those criteria are evaluated on the study area, especially since this analysis is done on aerial photos, while only low-quality images are provided in the article. | Activity 1 is further applied to comprehensively appraise the rock's exposed condition through the high-resolution DEM and aerial photo interpretation.
(3). Thank you for the correction, the "cliff" should be "scarp", which has been corrected in the overall manuscript, and the method is suitable for slow-moving slopes. The revised manuscript will add the Activity analysis process of slope unit No.11 from aerial photos (Fig.8) as an example to explain in detail.
(4). The images are provided with high quality in the revised manuscript. |
| Finally, the method is presented as "initially validated" by showing that a landslide occurred where the risk has been identified as the highest. The reason for the risk to be the highest in that slope unit is mainly that it is one of the few where there are elements at risk, although the hazard is relatively high as well. The susceptibility is medium and the activity high, but does the activity include the landslide that is described? Anyway, it is difficult to validate such a method and another aspect that I would like to be discussed is the applicability of the method on a more regional scale. It is thought to be fast, but I suspect that it is relatively time-consuming, and I would therefore like to read more about the context in which it could be used and how it would perform compared | Thank you for the suggestion.
(1). The landslide event in 2016 is not included in the Activity analysis.
(2). The proposed method intends to provide a rapid analysis according to the scoring indicators in the Tables after obtaining the initial geological, DEM, and aerial photo data. Indeed, identifying scarps in the activity analysis will take a little time because it is necessary to compare the evolution of possible features at different times for the exact location. Please refer to Fig.8 as an example.
(3). According to the official observations in the recent decade, the |

| | |
|---|---|
| to a more regional analysis. | Neikuihui tribe is the only one with a landslide case for validation. Although only one regional analysis is revealed in this study, the authors believe that the proposed rapid risk zoning process is ready to be applied in the next phase at other hillside tribes. |
| 1. I am not a native speaker, but I wonder if the word "Tribe" is really what you mean? | Thank you for the suggestion. The word "Tribe" is what we mean. |
| 2. l 33-67: there is a long list of articles, but I cannot really understand why they are listed here, how they relate to your work and what knowledge gap your study intends to fill. | Thank you for the suggestion. The paragraph collects significant risk analysis findings and the corresponding applications to reveal the basics of the risk analysis research method. For example, Varnes et al. provided the risk assessment principle in 1984 as Risk = Hazard × Exposure × Vulnerability; Corominas and Mavrouli (2011) stated a completed deep-seated landslide risk assessment must include Susceptibility, Hazard, and Vulnerability. The authors believe these are important and relevant documents closely related to this study. |
| 3. l 89-93: the description of the area is quite confusing. For example, the last sentence states that "most residents have moved north…" but where were they coming from? And does Neikuihui belong to the Kuihui village? | Thank you for the suggestion. The context has been revised: "They are the inhabitants who lived here before, named aborigines in Taiwan. Since the Neikuihui tribe has only one external road and frequent rockfall disasters, most residents have moved north to Kuihui Village, and only about 15 households are left in the Neikuihui tribe." |

| | | |
|---|---|---|
| 4. | l 102 (and figure 2): you present two different formation, but the rock type is mentioned only for the Tatongshan/Tatungshan formation. Further details on the Aoti formation would be useful. | Thanks for the suggestion. The text has been edited: "Figure 2e is a 1:25,000 geological map of the Central Geological Survey (2020). The strata include the Tatongshan and the Aoti formations, of which the Tatongshan formation is composed of black hard shale and siltstone interbedded, often forming steep slopes along the river bed. Aoti formation is composed of sandstone with a coal seam." |
| 5. | Figure 1: the map to the left lacks a scale and the north arrow does not corresponds to the image on the right since its top points to the south. Also, the quality of the image on the right is not sufficient to understand what we are looking at. A map with buildings and roads would help the reader. I am also wondering if it is a perspective view. | Thanks for the suggestion. Fig. 1 adds a global coordinate to the main map of Taiwan and provides a compass and road lines corresponding to the Neikuihui tribe area. We modified those perspective views or 3D maps updated with DEM and high quality aerial photos for better visualization. |
| 6. | Figure 2: I assume this one is a perspective view. I am disappointed by the quality since you apparently have a 1m-DEM. Could you improve the quality and provide a hillshade? Also I don't think that the CS-map is very helpful | Thank you for the suggestion. Fig. 2 is updated with high quality, and the hillshade map is also provided as a base map in Fig.2f. Since CS-map is provided for the comparison of the Relief map, we also update the quality for better visualization. |
| 7. | Figure 4: The sizes of the slope units are very different. I wonder how it impacts the analysis. | Thank you for the suggestion. When delimiting sloping land units, the authors utilized slope aspect, slope, ridgeline, river valley line, and geology for analysis, leading to the different area sizes of the slope |

| | | units. The authors spent much time examining the activity of the large slope unit through the aerial photo and DEM, as mentioned in the previous response. |
|---|---|---|
| 8. | Table 4: I don't understand the "raw grades" and "adjusted grades" and why for example the class with "less than one household" (isn't that 0?) gets a raw grade of 3 and an adjusted grade of 9 when it should be 0 if there is no household. "Households 3 to 4" and "Households 1 to 2" should be renamed "3 to 4 households" and "1 to 2 households" respectively. "More than five households" should be renamed "5 or more households". The summation of grades gives a "low" level from 1 to 11, but if "less than one household" gives 9 points, then it can't be below 9. I think the exposure level should be 0 if there is nothing otherwise the method gives an "extremely low risk" when there is actually no risk. | Your statement is correct, and we have modified the raw grade of the case "Less than 1 household" to one in Table 9, as well as the interpretation results.

According to the standard practices on risk management as proposed by previous kinds of literature, the authors divide the summation of grades in analysis into three levels (low, medium, and high) or five levels (extremely low, low, medium, high, and extremely high). |
| 9. | Vulnerability analysis: the vulnerability analysis is based on the distance to the landslide, but how are the landslide source, run-out and deposition areas defined? | Thank you for the suggestion. The authors add the vulnerability analysis process of slope unit No.11 as an example in the text to explain in detail. |
| 10. | l 217: The citation refers to an article describing a vulnerability | Thank you for the correction. Previously, we mainly quoted the |

| | |
|---|---|
| index that is combining several criteria including the physical properties of the buildings. I do not see the similitude with the method you are using. | vulnerability index formula mentioned in the article of Papathoma-Köhle et al. (2019). Based on your suggestion, we replaced it with the literature of Papathoma-Köhle et al. (2017), which is closer to the research content. |
| 11. Table 7: Slope unit 4 is considered a dip-slope, but only a tiny portion of it is inside the dip-slope polygon. I wonder if the polygon of dip-slope has been drawn at an appropriate scale for this analysis. Otherwise, is the slope unit well-defined? Or is the rest of the slope unit not a dip-slope because of a geological folding? | Thank you for the suggestion. The range and location of the dip slope at slope unit 4 and 5 are from the 1/25,000-scale official distribution maps of geologically sensitive areas provided by the Geological Survey of the Ministry of Economic Affairs of Taiwan. Despite the range and location of the dip slope being rough as mapping to the slope unit 4, this study preliminarily assumes the slope unit 4 as a dip slope. |
| 12. Table 12: There are 9 households in unit 5, but 1-2 according to table 11… which one is wrong? | Thank you for the correction. The authors have corrected the number of households in No.5. in Table 12, and changed the total score and figures accordingly. |
| 13. l 341: what is the residential house No. 8. Do you mean a house in slope unit No. 8? | Thank you for the correction. We removed the word "No.8". |
| 14. l 395: JSPJ is mentioned in the author contributions, but is not a co-author | Thank you for the suggestion. This part is wrongly planted and has been modified to CCC. |

New citation list:

1. Alvioli, M., Marchesini, P., Reichenbach, M., Rossi, F., Ardizzone, F., and Fiorucci, F.: Automatic delineation of geomorphological slope units with slope units v1.0 and their optimization for landslide susceptibility modelling Geosci. Model Dev., 9 (2016), 3975-3991, 10.5194/gmd-9-3975-, 2016.

2. Alvioli, M.,Guzzetti, F., and Marchesini, I.: Parameter-free delineation of slope units and terrain subdivision of Italy, Geomorphology., 358, https://doi.org/10.1016/j.geomorph.2020.107124, 2020.

3. Carrara, A.: Multivariare models for landslide hazard evaluation Math. Geol., 15 (3), 403-426, 10.1007/bf01031290, 1983.

4. Carrara, M., Cardinali, R., Detti, F., Guzzetti, V., and Pasqui, P.: Reichenbach GIS techniques and statistical models in evaluating landslide hazard Earth Surf. Process. Landf., 16 (5) (1991), 427-445, 10.1002/esp.3290160505, 1991.

5. Carrara, A., Guzzetti, F.: Geographical Information Systems., Kluwer Academic Publisher, Dordrecht, The Netherlands June 1995, 342, ISBN-13: 9780792335023, 1995.

6. Guzzetti, A., Carrara, M., and Cardinali, P.: Reichenbach Landslide hazard evaluation: a review of current techniques and their application in a multi-scale study, Central Italy, Geomorphology, 31, 181-216, 10.1016/s0169-555x(99)00078-1, 1999.

7. Guzzetti, F.: Landslide Hazard and Risk Assessment, PhD Thesis, Mathematics Scientific Faculty, University of Bonn, Bonn, Germany, https://nbn-resolving.org/urn:nbn:de:hbz:5N-08175, 2005.

8. Guzzetti, A.C., Mondini, M., Cardinali, F., Fiorucci, M., and Santangelo, K.T.: Chang Landslide inventory maps: new tools for an old problem, Earth-Sci. Rev., 112 (1), 42-66, 10.1016/j.earscirev.2012.02.001, 2012.

9. Malamud, D.L., Turcotte, F., and Guzzetti, P.: Reichenbach Landslide inventories and their statistical properties Earth Surf. Process, Landf., 29 (6), 687-711, 10.1002/esp.1064, 2004.

10. Papathoma-Köhle, M., Gems, B., Sturm, M. and Fuchs, S.: Matrices, curves and indicators: a review of approaches to assess physical vulnerability to debris flows. Earth-Sci, Rev., 171, 272–288, https://doi.org/10.1016/j.earscirev.2017.06.007, 2017.

11. Papathoma-Köhle, M., Schlögl, M., Dosser, L., Roesch, F., Borga, M., Erlicher, M., Keiler, M., and Fuchs, S.: Physical vulnerability to dynamic flooding: Vulnerability curves and vulnerability indices. J. Hydro., 607, 127501, https://doi.org/10.1016/j.jhydrol.2022.127501, 2022.

Removed citation list:

1. Guzzetti, F.: Landslide Hazard and Risk Assessment, PhD Thesis, Mathematics Scientific Faculty, University of Bonn, Bonn, Germany, https://nbn-resolving.org/urn:nbn:de:hbz:5N-08175, 2005.

2. Papathoma-Köhle, M., Schlögl, M., and Fuchs, S.: Vulnerability indicators for natural hazards: an innovative selection and weighting approach, Sci. Rep., 9, https://doi.org/10.1038/s41598-019-50257-2, 2019.

3. Xiong, J.N., Sun, M., Zhang, H., Cheng, W., Yang, Y.H., Sun, M.Y., Cao, Y.F., and Wang, J.: Application of the Levenburg–Marquardt back propagation neural network approach for landslide risk assessments, Nat. Hazards Earth Syst. Sci., 19, 629–653, https://doi.org/10.5194/nhess-19-629-2019, 2019.

---

## Author Response (AR2)

Paper Title: " Rapid Landslide Risk Zoning toward Multi-Slope Units around the Neikuihui Tribe for Preliminary Disaster Management "

**Reviewer #1:**

| Comment | Author's Response |
|---|---|
| Thanks to the authors for answering all comments raised in the first round of reviews in an adequate manner. Please note that line numbers refer to the version containing authors* tracked changes.

I do have some additional comments, which need further clarification. Many of them are of a merely technical nature. However, I do advise to revise parts of the discussion section to be more precise and explicit about benefits and limitations of the method. | Authors appreciate the comments and carefully examine the manuscript for the suggestions as follows: |
| 1. l.16: "assess" or "integrate" instead of "apply the susceptibility, ..." | 1. "assess" is used as suggested. |
| 2. l.94: To whom does "They are" refer to in this sentence? This is unclear, please rephrase. | 2. We rephrase the sentence as "The most residents are aborigines in Taiwan." |
| 3. l.98: I suggest to use " basis" instead of "basement" | 3. "basis" is used instead of "basement". |
| 4. l.100: I'm wondering if it is sufficiently clear to a wider audience, what a "CS map" is. I suggest add a half sentence explaining the actual type of output in this type of topographic map. See also the next point. | 4. Asahi (2014) proposed a CS (Curvature-Slope) map which can be made of altitude, curvature, and slope degree. Fig. 2c is provided with color attributes suggested by Asahi (2014) that Light Blue indicates valleys and Light Red indicates the ridge. |
| 5. Figure 2 (c) and (d): A continuous scale legend is somewhat | 5. The ranges between ridge and valley is added as suggested. The |

unorthodox for discrete features ranging from "Ridge" to "Valley". Using Ridge and Valley is fine, but I suggest to also add the actually plotted variables (i.e. their ranges) there. Also, if Figs. (c) and (d) are included for comparative purposes, wouldn't it be better to use the same color palette?

6. l.146: "Martinello et al. (2020) brought the Imera Settentrionale watershed in northern Sicily, Italy, as the research scope ..." is a somewhat strange sentence. Why not just "Based on an analysis of the Imera Settentrionale watershed in northern Sicily, Italy, Martinello et al. (2020) found slope units to be superior for representing landslide susceptibility as, as opposed to (I suppose: grid cells) due to (reason here)".

7. l.186: lowercase s in susceptibility

8. l.188: I would replace "The statistical one uses numerous landslides to analyze the corresponding unstable slopes." with "Statistical approaches require representative landslide inventories as training data set which are used to characterize the corresponding unstable slopes."

9. - l. 188f: "Factors such as topographical and geological conditions were marked with weights and rankings statistically, leading to objective results in practice." This sentence is unclear. Is this a general description following up on the description of statistical methods? Than the tense should be present tense. Or was this done

authors suggest keep the original color attributes of CS and RED maps since the main comparative purpose is trying to identify the similar topographic feathers from different maps.

6. The sentence is modified as "Based on an analysis of the Imera Settentrionale watershed in northern Sicily, Italy, Martinello et al. (2020) found slope units to be superior for representing landslide susceptibility as a real spatial scale in geomorphological form."

7. Modified as suggested.

8. Modified as suggested.

9. The sentence is modified as "Factors such as topographical and geological conditions are marked with weights and rankings in the statistical method, leading to objective results in practice."

| | |
|---|---|
| in this study? If yes, how? | 10. Modified as suggested. |
| 10. - l. 191: lowercase s in susceptibility | 11. The sentence is modified as "Then susceptibility of the landslide was evaluated with the Logistic regression, resulting in main factors of the slope degree, lithology, and dip slope, as well as the adjacent conditions to a river and fault." |
| 11. - l. 191: please state "using logistic regression" explicitly, the term "logistically" is incorrect in this context | |
| 12. - l.192: Does "adjacent conditions" mean that the features "distance to rivers" and "distance to faults" were considered as independent variables in the regression model? | 12. Yes, these factors are summarized in Table 1. |
| 13. - l. 194: "Afterward, regional statistical results were applied to calculate the occurrence index normalized from 1 to 2." I cannot really follow what was done here. Were the prediction results of the logistic regression (i.e. values in the interval [0,1]) transformed into the range [1,2]? If so, why is this a normalization? Normalization usually refers to a transformation into a standard normal distribution with mean zero and sd = 1. | 13. Authors sorry for the wrong word "normalized". Actually, the Occurrence index is defined in Forest Bureau (2013), then Forest Bureau (2013) proposed the grade results as listed in Table 1 |
| | 14. Authors would like to point out that this study assumes factors from the logistic regression model as the indicators. Thus, there is no differentiation between the susceptibility map and the indicator-based approach as mentioned by the reviewer. |
| 14. - Upon reading this previous paragraph (Beginning of Section 3.2.1) a couple of times I think that I am still slightly confused here, since to me there is no clear differentiation between the susceptibility map (based on the logistic regression model) and the indicator-based approach. I suggest to streamline this section a bit to avoid misunderstandings here. | 15. Modified as suggested. |
| 15. - 205ff: I would write activity with a lowercase a. | 16. Yes, the image includes a NIR band rather than being a standard VIS orthophoto. |
| 16. - Figure 8: This image seems to include a NIR band rather than being a standard VIS orthophoto? | |

| | | | |
|---|---|---|---|
| 17. | - Figure 11: Is there a slope unit in light green ("extremely low")? (This is hard to see in the version provided for review). If not, this category can be removed from the legend for clarity. | 17. | No. 1, 2, 3, 6, 7 ,8 9, 10, and 12 slope unites in Fig. 11 are rated in "extremely low", and we modified the color for better visualization as suggested. For consistency, Fig. 9, 12 are modified as well. |
| 18. | - 461: what is "semi-experienced statistical prediction"? | 18. | The term of "semi-experienced is removed to avoid confusing. |
| 19. | - 464: "the quantitative method is introduced" this is unclear. where or to whom is it introduced, and why is the introduction relevant? | 19. | The sentence is modified as "The quantitative method requires very detailed site information and statistics of various parameters, often taking a lot of time and cost." |
| 20. | - 466: "A qualitative description does not need to quantify each factor, which is described hierarchically". What is described hierarchically? Please be more explicit. | 20. | The sentence is modified as "A qualitative description does not need to quantify each factor, which is described hierarchically by assessing hazard (including susceptibility and activity levels), exposure, and vulnerability as proposed in this study." |
| 21. | - "Therefore, the initial risk assessment is more suitable with qualitative descriptions." I would rather say it is easier to accomplish, not necessarily more suitable. | 21. | Modified as suggested. |
| 22. | - 470: "The Risk zoning framework designed in this study includes the Activity assessment when grading the Hazard, which can make up for the lack of time change in the Susceptibility assessment and represent the actual site activities. However, it requires little time to analyze visually." (1) What do you mean with "lack of time change in the Susceptibility assessment"? (2) What does "it requires little time to analyse visually?" mean? It requires little time would mean that this is very fast, but I am under the impression that this is rather a drawback, and that some effort is required for this analysis? | 22. | The paragraph is modified as "The Risk zoning framework designed in this study includes the Activity assessment when grading the Hazard, which examines a possible surface change of a slope at different times to represent the actual site activities. In contrast, the Susceptibility assessment considers general environmental factors as revealed in Table 1. However, the Activity assessment requires little effort to analyze the ground feathers visually from the aerial images, but still faster than a quantitative method from previous experience." |

| | |
|---|---|
| 23. - l. 479: "the economic conditions are disadvantaged" does not seem to be proper English to me.

24. - Again, dropbox is not the optimal place to store research data. Please deposit the data on a proper data repository. | 23. The sentence is modified as "However, there are still households with underprivileged groups and economic weakness in this area."

24. Authors used the Zenodo DB for data repository as suggested. (https://zenodo.org/record/6513416#.YnDoHNpByUk) |

---

## Author Response (AR3)

Dear Authors

Thanks you for the careful overworking of your manuscript. I am happy to accept it for publication in NHESS. However please make sure that your figures have adequate quality for publication.

Response:

Authors prepare and upload the stand-alone figures with adequate quality for publication.